# Sperm Quality Affected by Naturally Occurring Chemical Elements in Bull Seminal Plasma

**DOI:** 10.3390/antiox11091796

**Published:** 2022-09-12

**Authors:** Filip Tirpák, Marko Halo, Marián Tomka, Tomáš Slanina, Katarína Tokárová, Martyna Błaszczyk-Altman, Lucia Dianová, Peter Ivanič, Róbert Kirchner, Agnieszka Greń, Norbert Lukáč, Peter Massányi

**Affiliations:** 1AgroBioTech Research Centre, Slovak University of Agriculture in Nitra, Tr. A. Hlinku 2, 949 76 Nitra, Slovakia; 2Institute of Biotechnology, Faculty of Biotechnology and Food Science, Slovak University of Agriculture in Nitra, Tr. A. Hlinku 2, 949 76 Nitra, Slovakia; 3Institute of Applied Biology, Faculty of Biotechnology and Food Science, Slovak University of Agriculture in Nitra, Tr. A. Hlinku 2, 949 76 Nitra, Slovakia; 4Institute of Biology, Pedagogical University of Krakow, Podchorazych 2, 30-084 Krakow, Poland; 5Slovak Biological Services, Kremnička 2, 974 05 Banská Bystrica, Slovakia

**Keywords:** seminal plasma, spermatozoa, bovine, RedOx status, motility, viability, microelements, macroelements, heavy metals

## Abstract

This study monitored the chemical and biochemical composition of bovine seminal plasma (SP). Freshly ejaculated semen (*n* = 20) was aliquoted into two parts. The first aliquot was immediately assessed to determine the sperm motion parameters. Another motility measurement was performed following an hour-long co-incubation of spermatozoa with SP at 6 °C. The other aliquot was processed to obtain the SP. Seminal plasma underwent the analyses of chemical composition and quantification of selected proteins, lipids and RedOx markers. Determined concentrations of observed parameters served as input data to correlation analyses where associations between micro and macro elements and RedOx markers were observed. Significant correlations of total oxidant status were found with the content of Cu and Mg. Further significant correlations of glutathione peroxidase were detected in relation to Fe and Hg. Furthermore, associations of chemical elements and RedOx markers and spermatozoa quality parameters were monitored. The most notable correlations indicate beneficial effects of seminal Fe on motility and Mg on velocity and viability of spermatozoa. On the contrary, negative correlations were registered between Zn and sperm velocity and seminal cholesterol content and motility. Our findings imply that seminal plasma has a prospective to be developed as the potential biomarker of bull reproductive health.

## 1. Introduction

Intensive cattle breeding requires high fertility to be economically convenient. Even though the artificial insemination (AI) is frequently used, the presence of breeding bull in the herd is important to ensure the successful fertilization in the situations when the AI has not worked [1]. The significance of the fluid part of the semen is associated with both male and female reproductive systems [2]. Seminal proteins provoke an inflammatory reaction that is cleaning the intrauterine space and preparing the uterus for embryo implantation [3]. Proteins contained in SP are often bound up with cholesterol of sperm lipid membrane. Cholesterol is effluxed from the spermatozoa particularly during sperm capacitation. Fatty acids contribute to sperm energy production as an energy substrate designated for β-oxidation in sperm mitochondria [4,5]. The role of SP during natural mating is crucial due to its rich content of biomolecules and micro and macro elements which all together enable spermatozoa to succeed—to fertilize [6].

Seminal micro and macro elements directly alternate the sperm quality [7]. Minerals such as calcium and magnesium are appropriate examples of essential seminal chemical elements. Calcium regulates numerous physiological processes in spermatozoa such as spermatozoa maturation, acrosome reaction and motility [8,9]. Magnesium is engaged in enzymatic reactions in several pathways of energy metabolism and biosynthesis of nucleic acids. However, the levels of these essential elements must not exceed certain levels. Enhanced Mg concentration in SP is associated with premature ejaculation. Moreover, Mg antagonizes Ca on the intracellular level. Thus, it is not only the level but also the mutual ratio of seminal elements that matter [10,11,12]. Trace elements such as Zn, Se, Fe, Cu, etc. reduce the common bias concerning the exclusively harmful effect of heavy metals. In trace amounts, they are involved in basic physiological and biochemical processes [13]. The importance of these elements for the appropriate functioning of the male reproductive system has been widely discussed and proved [14,15,16].

Male reproductive system is sensitive to exogenous effects and thus reflects the state of the environment, nutrition, etc. As the seminal plasma serves the spermatozoa as a transport and nourishing medium in the female reproductive tract, contamination or disbalance in constituents of SP may compromise the fertilization. Elevated levels of heavy metals in seminal plasma could be a result of chronic or acute exposure of the individual to heavy metal pollution [7]. The animal exposed to such an environment may undergo deleterious alternations of reproductive organs, resulting in the production of malformed gametes [17]. Spermatozoa specifically exhibit reduced sperm count, morphological defects (head, acrosome, mitochondrial segment, flagellum), DNA fragmentation, etc. [18].

The mode of action of heavy metal-induced modifications of reproductive system consists in excessive production of reactive oxygen species (ROS). Despite its irreplaceable function in proper physiological functioning, ROS is considered a major threat to spermatozoa. Pathophysiological imbalance between ROS and antioxidant activity of the SP, defined as oxidative stress (OS), most likely leads to detrimental changes in sperm morphology and physiology [19]. ROS play a crucial role in motility, capacitation, hyperactivation and acrosome reaction and thus very the same processes get affected by excessive ROS [20]. Spermatozoa are highly vulnerable to the ROS activity also due to its structure. The polyunsaturated acids contained in sperm membranes easily undergo lipid peroxidation [19].

SP possesses a very effective weapon to battle the excessive ROS—enzymatic antioxidants. Superoxide dismutase (SOD) and glutathione peroxidase (GPx) belong among the most abundant enzymatic antioxidants contained in SP [21]. SOD catalytically transforms superoxide to oxygen and hydrogen peroxide that is further converted by GPx into water [22].

The concept of using SP as a biomarker of reproductive health and reproductive toxicity has already been proposed in stallions [6,23,24] and boars [25,26,27,28]. Concentrations of micro and macroelements in bull SP seem to have a determinizing role for sperm cryotolerance [29]. It is also assumed that the composition of SP has a massive impact on conceptus development [30,31]. These reports underline the need for the complex examination of SP and its possible further implications in reproduction. Seminal plasma as relatively accessible fluid has the potential of becoming a regularly tested specimen before the beginning of every breeding season.

The main aim of the study was to quantify naturally occurring concentrations of selected chemical elements (Al, Ba, Ca, Cd, Co, Cr, Cu, Fe, Hg, K, Mg, Mn, Mo, Na, Ni, Pb, Sr, and Zn), lipids (cholesterol and triglycerides), proteins (total proteins and albumins), and to evaluate RedOx status (TOS, FRAP, SOD, GPx, and MDA). Moreover, statistical correlations imply interactions between SP composition, RedOx status, and sperm quality parameters.

## 2. Materials and Methods

### 2.1. Collection and Processing of Semen

Semen was acquired from 20 routinely collected breeding bulls at the age of 29–38 months. Bulls subjected to the study were of the following breeds: Black Holstein, Red Holstein and Slovak Simmental Cattle. All bulls were held under the same housing and feeding conditions throughout the experiment at a breeding station located in the District of Nitra (Slovakia). Semen was collected by the trained technician using the pre-warmed artificial vagina. The animals were meticulously handled in accordance with the ethical guidelines of the Slovak Animal Protection Regulation RD 377/12, conforming to European Union Regulation 2010/63. Immediately after the collection, the semen was split into two aliquots. The first aliquot was used for the spermatozoa quality assessment, and the other was processed to obtain supernatant by centrifugation (3000× *g* for 10 min). The supernatant was stored at −80 °C for the further analyses of seminal plasma (SP).

### 2.2. Motility Analysis

Spermatozoa motility was analyzed at time 0 and after an hour-long co-incubation with seminal plasma at 6 °C. Sample was transferred into a Makler Counting Chamber^®^ (Sefi–Medical Instruments, Haifa, Israel) and allowed to warm up for 15 s prior to analysis. Using the bull set-up, basic and supportive parameters were selected—total motility (MOT; %), progressively motility (PRO; %), velocity curved line (VCL; µm/s), amplitude of lateral head displacement (ALH; µm), and beat cross frequency (BCF; Hz) were evaluated using a CASA system—Sperm Vision^®^ program (Minitub, Tiefenbach, Germany) operated with a microscope (Olympus BX 51, Tokio, Japan). To carry out a single CASA assessment, seven different fields of view of Makler Counting Chamber were assessed as previously defined by Slanina et al. [32]. In order to characterize the period of the analysis, every mentioned parameter is labeled with an underscore (_) and the time of the analysis (0—initial measurement; 1—measurement after an hour incubation) (e.g., MOT_0, MOT_1, etc.) [6].

### 2.3. Mitochondrial Activity

Sperm viability was determined by the mitochondrial test of toxicity (MTT). This colorimetric assay observes the conversion of 3-(4,5-dimetylthiazol-2-yl)-2,5-diphenyltetrazolium bromide (Sigma-Aldrich, St. Louis, MI, USA) to purple formosan particles by the activity of succinate dehydrogenase produced by viable mitochondria. Optimal density was evaluated at a wavelength of 570 against 620 nm by a Multiskan FC spectrophotometer (ThermoFisher Scientific, Vantaa, Finland). Acquired data were expressed as absorbance (Abs) [6].

### 2.4. Detection of Essential and Heavy Metals in Seminal Plasma

All the chemicals used in the process of sample preparation were of high purity. The samples (0.5 mL) were digested in the high-performance microwave digestion system Ethos UP (Milestone Srl, Sorisole, BG, Italy) in a mixture of 5 mL HNO_3_ (TraceSELECT^®^, Honeywell Fluka, Morris Plains, NJ, USA) and 1 mL of H_2_O_2_ (30% for trace analysis, Merck, Darmstadt, Germany). All samples, along with the blank sample, were mineralized in accordance with the manufacturer’s recommendation. The method involves heating and cooling stages. During the heating phase, the samples were continuously heating up to 200 °C for 15 min, and this temperature was retained for another 15 min. Afterward, the samples were subjected to 15 min of active cooling to reach the temperature of 50 °C. The digests were filtered through the Sartorius filter discs (grade 390) (Sartorius AG, Goettingen, Germany) into the volumetric flasks and filled up with ultrapure water to a volume of 50 mL or further diluted if necessary. The dilution was taken into consideration during the final processing of results [33].

Quantification of the chemical elements (Al, Ba, Ca, Cd, Co, Cr, Cu, Fe, K, Mg, Mn, Mo, Na, Ni, Pb, Sr, and Zn) present in bull seminal plasma was conducted using inductively coupled plasma—optical emission spectrophotometer (ICP Thermo iCAP 7000 Dual; Thermo Fisher Scientific, Waltham, MA, USA). Multielement standard solution V for ICP (Sigma-Aldrich Production GmbH, Switzerland) was used for calibration. Obtained results were expressed as g/kg, mg/kg or μg/kg.

Among other heavy metals, total mercury concentration (Hg) was also assessed; however, a different methodical approach was used. Each sample (100 µL) was analyzed by a cold vapor atomic absorption spectrometer MA−2 (Nippon Instruments Corporation, Bukit Batok, Singapore), while no pre-analytical procedures were needed. The mean value of two replicates was used for further analyses if the RSD between replicates was less than 10%; otherwise, the sample was reanalyzed [34]. The limits of quantification (LOQs) for all assessed chemical elements are summarized in Table 1.

### 2.5. Ferric Reducing Ability of Plasma (FRAP)

The FRAP analysis was carried out based on the Benzie and Strain [35], modified by Tvrda [36]. The FRAP reagent contains 10 mmol/L TPTZ (2,4,6- tripyridyl-s-triazine; Sigma-Aldrich, St. Louis, MO, USA) solution in 40 mmol/L HCl (Centralchem, Bratislava, Slovakia) and 5 mL of 20 mmol/L ferric chloride (Centralchem, Bratislava, Slovakia) and 50 mL of 0.3 mmol/L acetate buffer (pH = 3.6; Centralchem, Bratislava, Slovakia). Samples (100 μL) were mixed with a 3 mL FRAP reagent, and the absorbance of the reaction mixture was evaluated at 593 nm using a Multiskan FC spectrophotometer (ThermoFisher Scientific, Vantaa, Finland). The absorbance reading was repeated after 4 minutes of co-incubation of samples/standards with the reaction solution.

### 2.6. Total Oxidant Status (TOS)

The TOS analysis is based on the oxidation of ferrous ions-o-dianisidine complexes to ferric ions. This process is regulated by the oxidants present in the sample. The whole analysis is based on the two reaction solutions: TOS R1 and TOS R2. The TOS R1 is composed of 150 μmol xylenol orange disodium salt (Sigma-Aldrich, St. Louis, MO, USA), 140 mmol sodium chloride (Sigma-Aldrich, St. Louis, MO, USA), and 1.35 mol glycerol (Centralchem, Bratislava, SR) in 25 mmol H_2_SO_4_ (Sigma-Aldrich, St. Louis, MO, USA). The TOS R2 contains 5 mmol ferrous ammonium sulfate hexahydrate (Centralchem, Bratislava, Slovakia), and 10 mmol *o*-dianisidine dihydrochloride (Sigma-Aldrich, St. Louis, MO, USA) in 25 mmol sulfuric acid (Sigma-Aldrich, St. Louis, MO, USA). Standards (H_2_O_2_) and samples of SP (35 μL) were pipetted in doubles to 96-well plate. TOS R1 (225 μL) was added, and the reference reading at 560 nm was performed using a Glomax Multi+ Detection System plate reader (Promega, Madison, Wisconsin, USA). After 10-min incubation, 11 μL TOS R2 was added to each well. Following a 3-min incubation period, the absorbance was spectrophotometrically assessed at the same wavelength. The assay was calibrated using hydrogen peroxide, and the results are expressed as μmol H_2_O_2_ Eq/g of protein [6,37].

### 2.7. Superoxide Dismutase (SOD)

SOD was analyzed on Randox RX Monza (Randox Laboratories, Crumlin, UK) using a RANDOX assay kit RANSOD (Randox Laboratories, Crumlin, UK) following the manufacturer’s instructions. SOD was spectrophotometrically quantified at 505 nm using Multiskan FC (ThermoFisher Scientific, Finland) and expressed as U/mg of total protein.

### 2.8. Glutathione Peroxidase (GPx)

Activity of glutathione peroxidase was analyzed using commercially available kit RANSEL (Randox Laboratories, Crumlin, UK) and analyzer Randox RX Monza (Randox Laboratories, Crumlin, UK). The decrease in absorbance is measured at 340 nm. Enzyme activity was expressed as U/mg of total protein.

### 2.9. Lipid Peroxidation (LPO)

Lipid peroxidation was assessed via the measurement of malondialdehyde (MDA) production employing the TBARS assay, adapted for a 96-well plate and ELISA reader. Samples were treated with 5% sodium dodecyl sulfate (SDS; Sigma-Aldrich, St. Louis, MO, USA) and exposed to 0.53% thiobarbituric acid (TBA; Sigma-Aldrich, St. Louis, MO, USA) dissolved in 20% acetic acid adjusted with NaOH (Centralchem, Bratislava, Slovakia) to pH 3.5, and afterwards boiled at 90–100 °C for an hour. Subsequently, the samples were placed on ice for 10 min to stop the reaction. The samples were subjected to centrifugation (1750× *g* for 10 min), and the obtained supernatant was analyzed for the concentration of MDA. The analysis was performed on Multiskan FC microplate photometer (Thermo Fisher Scientific Inc., Waltham, MA, USA) at 530–540 nm [38]. The content of MDA is expressed as μmol/g protein.

### 2.10. Biochemical Analysis of Seminal Plasma

Determination of biochemical parameters’ total proteins (TP), albumins (Alb), cholesterol (Chol), and triacylglycerides (TAG) was executed using DiaSys commercial kits (Diagnostic Systems GmbH, Holzheim, Germany). The measurements were performed employing Randox RX Monza analyzer (Randox Laboratories, Crumlin, UK) in accordance with manufacturer’s recommendations.

### 2.11. Statistical Analysis

The complete set of statistical analyses was performed using GraphPad 8 software (GraphPad Software Inc., San Diego, CA, USA). In terms of descriptive analysis, the following calculations were carried out: mean, standard deviation (SD), minimum values (Min), and maximum values (Max). All obtained data were examined for normal Gaussian distribution via the D’Agostino–Pearson normality test and Shapiro–Wilk normality test. Pearson correlation between markers of oxidative stress and chemical elements was performed to suggest the possible mechanism of action. Furthermore, the association between sperm properties and constituents of seminal plasma was assessed using Pearson correlation. Significance was defined at *p* < 0.05 (a) and at *p* < 0.01 (A). Heatmaps with clustering were prepared to visualize interactions (Pearson correlations coefficients—*r*) of chemical elements and markers of oxidative stress on sperm properties [6].

## 3. Results

### 3.1. Sperm Quality

The percentage of motile spermatozoa was in fresh semen 90.64%. One-hour storage of cooled spermatozoa caused a reduction of motility to 69.45%. Despite the decrease of motility, examined semen showed an impressive ratio of progressively motile cells, starting at 86.24% in the initial period and reaching 62.81% in the second measurement. Regarding velocity, relatively stable values of VCL were recorded throughout the incubation. The same feature applies to BCF and ALH parameters. Differences in mitochondrial activity were monitored on the scale from 0.03 Abs to 0.46 Abs (Table 2).

### 3.2. Markers of Oxidative Stress in Seminal Plasma

As demonstrated by the TOS and FRAP, variability in redox status was obvious, which was eventually mirrored at the level of LPO quantified via MDA. The activity of SOD was higher than the activity of GPx, suggesting the higher incidence of superoxide anion in the SP (Table 3).

### 3.3. Biochemical Composition of Bovine Seminal Plasma

Seminal plasma contained on average 49.03 g/L of proteins. Almost a third of the TP was represented by Alb (16.69 g/L). The concentration of cholesterol was determined at 1.81 mmol/L. Comparatively, the content of TAG in bull SP was 1.10 mmol/L (Table 4).

### 3.4. Chemical Composition of Bovine Seminal Plasma

The most abundant element detected in bull SP was K (3.27 g/kg) followed by Na and Ca present in much lower concentrations (1.31 g/kg and 330.32 mg/kg, respectively). The lowest values were detected assessing the Hg content. Values of observed chemical elements were sorted in descending order and implied the following arrangement: K > Na > Ca > Mg > Zn > Ni > Fe > Cr > Cd > Mo > Pb > Al > Cu > Ba > Co > Sr > Mn > Hg (Table 5).

Negative correlations between OS markers and metals were observed (Figure 1, Appendix A). Nevertheless, only interactions of Fe and Hg with GPx were significant (*p* < 0.05). On the contrary, a positive effect was monitored between oxidative status and Cu (*p* < 0.01) and Mg (*p* < 0.05).

Evaluating the correlation analysis, the association of cholesterol with several spermatozoa motility traits was found. The negative relationship of Chol with MOT_0 and PRO_0 differed in the significance as displayed in Figure 2 and Appendix A. Oppositely, positive interaction was detected with BCF_1 on the level of significance *p* < 0.05. Toxic metals present in the semen negatively correlated with total and progressive motility, although they lacked the statistical significance. On the contrary, Fe significantly (*p* < 0.05) stimulated total and progressive motility in the second measured interval. The strong negative correlation (*p* < 0.01) of Zn with VCL parameters indicates the harmful effect of zinc on spermatozoa velocity. With respect to biogenic elements, Mg positively correlated (*p* < 0.05) with VCL_0 and mitochondrial activity. The amplitude of lateral head displacement in the initial measurement was positively affected by Ca.

## 4. Discussion

Bull seminal plasma is considered an important fluid offering a source of antioxidant agents as well as a source of energy for spermatozoa. Notwithstanding the aforementioned, the composition of seminal plasma may be affected by the toxic action of environmental contaminants and thus impair the spermatozoa functionality [6,7].

Chemical elements of the current study occurred in the bovine SP in the following order of concentrations: K > Na > Ca > Mg > Zn > Ni > Fe > Cr > Cd > Mo > Pb > Al > Cu > Ba > Co > Sr > Mn > Hg. The results of the present study identify Cu as the main contributor to the generation of OS in examined samples. Copper analyzed in concentration 0.43 mg/kg was higher than 0.27 mg/kg mentioned by Knazicka et al. [39] and 0.057 mg/kg declared by Aguiar et al. [40] but lower than 0.71 mg/kg reported by Skiba and Gou [41] and 1.55 mg/kg reported by Tvrda et al. [42]. Even despite lower concentrations of Cu in SP, all authors mention the adverse effect on spermatozoa cells including decreased motility and viability and higher incidence of disrupted acrosomes. The disturbing effect of Cu was also proven in in vitro conditions [43,44,45]. Concerning redox homeostasis of sperm cells, we analyzed nonenzymatic parameters as well as enzymatic parameters including GPx and SOD. The activity of SOD is directly associated with Cu, Zn, or Mn [46]. Based on our results and further correlation analyses, we can confirm that the activity of SOD was not affected by the presence of measured concentration of chemical elements. However, an enhanced level of Cu may act as a prooxidant and thus induce generation of oxidative stress, which is seen in our results as a higher level of TOS (concentration of H_2_O_2_ in sample). Fe and Hg present in the samples may negatively affect redox balance in terms of decreased GPx activity, which is eventually verified by an elevated level of TOS. Both Fe and Hg act as prooxidants, while Fe is engaged in Fenton reaction, resulting in an increased concentration of oxygen radicals; Hg affects disulfide bonds of glutathione (GSH), which impairs the overall cycle of GPx and GSH. As stated by Muino-Blanco et al. [47], an elevated activity of GPx and SOD in bull SP indicates an effort to overcome the negative activity of ROS. The results obtained in the current study represent activities 1.65 U/g TP for GPx and 0.15 U/mg TP for SOD. The effect of Hg on ROS generation was proclaimed also by Arabi [48]. Moreover, Mukherjee et al. [49] describe Hg as an initiator of numerous spermatozoa alterations, including decreased sperm count, higher incidence of morphologically defective spermatozoa, and an increased ROS level and LPO. An elevated concentration of Ni in bull semen may be accompanied by pathological modifications of spermatozoa morphology [50]. Nickel also interferes with intracellular Ca^2+^ signalling, leading to impaired motility and excessive OS [51]. However, despite the concentration of 2.50 mg/kg detected in the present work, no association was determined by performed correlation analysis. Zinc is crucial for male reproduction and its deficiency may lead to deteriorating alterations in spermatogenesis [29,49]. Janicki and Cygan-Szczegielniak [52] describe the positive association of seminal Zn with total and progressive motility in bulls. The present study did not notice such correlation; on the contrary, the correlation of Zn with VCL (*p* < 0.01) was estimated. Zinc-impaired spermatozoa motility traits have been already reported by Marzec-Wróblewska et al. [9]. Other metals (Al, Ba, Cd, Co, Cr Pb, and Sr) negatively correlated with MOT and PRO after an hour of coincubation with SP; however, the statistical significance was missing. The toxicity of some of these chemical elements was already discussed [38,42,53]. Iron was the only metal positively affecting MOT and PRO on a level of significance *p* < 0.05. The similar findings were monitored by Tvrdá et al. [54]. Possible explanation of the proposed correlation arises from the iron being part of the hydrogen peroxide catlalyzing enzyme—catalase. Moreover, as a part of mitoferrin, Fe has a role in mitochondrial dynamics [55]. Biogenic elements exhibited positive associations (*p* < 0.05) with spermatozoa quality parameters. Magnesium correlated with VCL_0 and MTT while calcium affected ALH_1. Magnesium affects enzymes involved in the mitochondrial metabolism as well as regulates mitochondrial Ca^2+^ transport [56]. This not only elucidates the positive correlation with viability of mitochondria but also offers a hypothesis that magnesium-mediated synthesis of ATP may result in the enhanced velocity of spermatozoa.

Proteins contained in the bull seminal plasma have important functions not only in relation to spermatozoa quality but also markedly contribute to successful fertilization by interacting with the female tract [57,58,59]. The concentration of TP (49.03 g/L) in SP was lower than 75.2 g/L stated by Tribulo et al. [57]. Albumins quantified in concentration 16.69 g/L did not statistically interact with spermatozoa motility traits; however, they reportedly absorb lipid peroxides and thus help to preserve the cell membrane and maintain its motility [47,60]. The detected concentration of Alb corresponds with the study of Tvrda et al. [42]. Changes in concentrations of TP, Alb, Chol, and TAG in boar seminal plasma were observed between different breeds. Total cholesterol and triacylglycerols were the most variable parameters and were suggested as plausible markers of boar semen quality [61]. In general, lipids are considered a basic structural unit of spermatozoa. In addition, cholesterol is engaged in essential physiological functions [5,62]. According to performed correlation analysis, cholesterol level negatively affected total (*p* < 0.01) and progressive motility (*p* < 0.05). Brinsko et al. [62] report low levels of seminal cholesterol in bulls with unexplained subfertility associating it with morphological alterations of head and midpiece of spermatozoa. Based on their findings, the ratio between cholesterol and phospholipids regulates capacitation and acrosome reaction. Beer-Ljubicic et al. [5] explain that the content of cholesterol in bull seminal plasma varies seasonally and depends on the age of the sire. Still, Chol level detected in the present study was higher. This indicates the possible noxiousness of SP Chol in the concentration of 1.81 mmol/l. On the contrary, research in humans describes cholesterol as a beneficial component of SP. Neergaard et al. [63] report the positive associations between SP Chol and sperm concentration, morphology, and motility, implying the involvement in spermatogenesis. This claim is supported by a high abundance of enzymes engaged in Chol metabolism in testicular cells. Triacylglycerols, another component of the SP lipid profile, play a role in the sperm metabolism and energy production. A few studies suggest a positive correlation between seminal TAG and sperm motility and fertilization ability [64,65]; however, findings of this work do not support this statement.

The findings of the current study must be seen in light of two major limitations. The first limitation concerns not generally established accepted reference values of the composition of seminal plasma. Thus, it is difficult to evaluate the quality of the SP. To compare obtained results, we used previously published data found in literature. We believe that the SP may serve as a potential indicator of reproductive health, mainly in correspondence to the environmental factors and the quality of the sire’s diet and thus should be investigated more. The secondary limitation to the generalization of obtained results is that the outcome of the study is rather theoretical and based on the statistical correlations. Nevertheless, the proposed associations have rational biological explanation, granting the obtained results a valuable mark of importance and plausible causality. Further research is required to endorse or refute our findings.

## 5. Conclusions

The presence of metals in the bull seminal plasma is in descending order according to their concentrations as follows: K > Na > Ca > Mg > Zn > Ni > Fe > Cr > Cd > Mo > Pb > Al > Cu > Ba > Co > Sr > Mn > Hg. Correlation analyses propose a possible link between Cu and Mg and enhanced TOS level, while Fe and Hg interacted with GPx activity. Interestingly, Fe and Mg positively correlated with sperm motility and velocity. Supposedly, cholesterol contained in bull SP interfered with total and progressive motility. A strong negative correlation between seminal Zn concentration and sperm velocity was registered. Considering the threat for spermatozoa issued by an imbalance of metals, proteins, lipids or impaired homeostasis in bovine SP, it is important not to forget to routinely monitor the content of the SP.

## Figures and Tables

**Figure 1 antioxidants-11-01796-f001:**
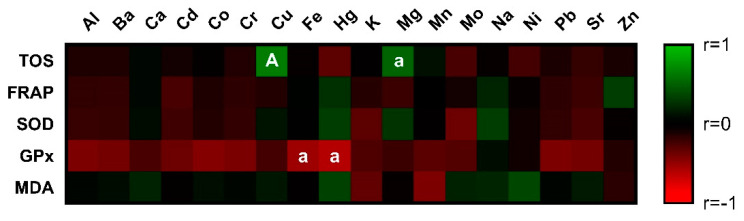
Correlations: chemical elements vs. markers of oxidative stress. The level of significance: a (*p* < 0.05); A (*p* < 0.01).

**Figure 2 antioxidants-11-01796-f002:**
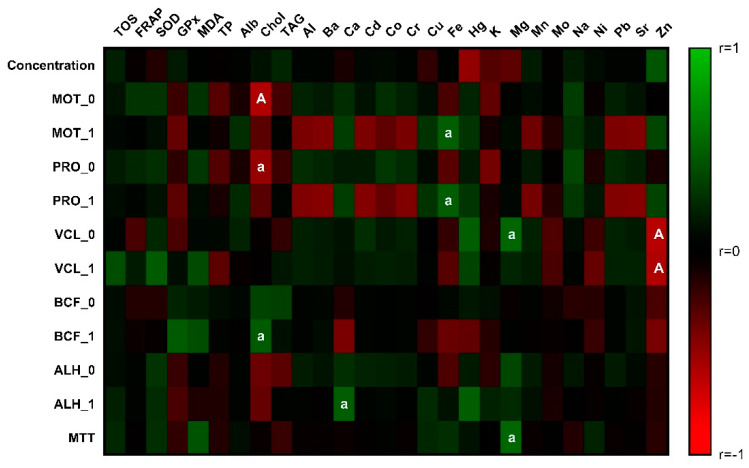
Correlations: spermatozoa parameters vs. markers of oxidative stress, biochemical parameters, and chemical elements. The level of significance: a (*p* < 0.05); A (*p* < 0.01).

**Table 1 antioxidants-11-01796-t001:** The LOQs for each monitored chemical element.

	LOQ	Unit
Al	0.0254	mg/kg
Ba	0.0110	mg/kg
Ca	0.0009	mg/kg
Cd	0.0012	mg/kg
Co	0.0060	mg/kg
Cr	0.0397	mg/kg
Cu	0.0110	mg/kg
Fe	0.0038	mg/kg
Hg	0.0200	μg/kg
K	2.0508	mg/kg
Mg	0.0013	mg/kg
Mn	0.0011	mg/kg
Mo	0.0115	mg/kg
Na	0.6116	mg/kg
Ni	0.0057	mg/kg
Pb	0.0332	mg/kg
Sr	0.0060	mg/kg
Zn	0.0331	mg/kg

LOQ—Limit of quantification.

**Table 2 antioxidants-11-01796-t002:** Bull sperm quality parameters.

	Mean	SD	Min	Max
Concentration (10^9^/mL)	2.26	0.80	0.63	3.96
MOT_0 (%)	90.64	6.04	77.98	96.66
MOT_1 (%)	69.45	18.87	37.99	94.31
PRO_0 (%)	86.24	6.87	69.26	93.58
PRO_1 (%)	62.81	18.99	32.75	88.17
VCL_0 (µm/s)	133.05	16.40	104.00	171.63
VCL_1 (µm/s)	124.27	23.40	97.21	206.96
BCF_0 (Hz)	31.10	3.34	27.22	38.78
BCF_1 (Hz)	29.06	3.15	25.24	34.99
ALH_0 (µm)	5.37	0.71	3.83	6.41
ALH_1 (µm)	5.30	0.69	3.96	6.70
MTT (Abs)	0.20	0.13	0.03	0.46

MOT_0—motility in initial time; MOT_1—motility following 1-h incubation at 6 °C; PRO_0—progressive motility in initial time; PRO_1—progressive motility following 1-h incubation at 6 °C; VCL_0—velocity curved line in initial time; VLC_1—velocity curved line following 1-h incubation at 6 °C; BCF_0—beat cross frequency in initial time; BCF_1—beat cross frequency following 1-h incubation at 6 °C; ALH_0—amplitude of lateral head displacement in initial time; ALH_1—amplitude of lateral head displacement following 1-h incubation at 6 °C; MTT—mitochondrial toxicity test following 1-h incubation at 6 °C.

**Table 3 antioxidants-11-01796-t003:** Markers of oxidative stress in bovine seminal plasma.

	Mean	SD	Min	Max
TOS (μmol H_2_O_2_/g TP)	0.23	0.15	0.06	0.59
FRAP (μmol Fe^2+^/g TP)	32.68	7.63	24.65	53.02
GPx (U/g TP)	*1.66*	*1.02*	*0.62*	*4.73*
SOD (U/mg TP)	0.15	0.04	0.09	0.27
MDA (µmol/g TP)	0.92	0.80	0.01	2.80

TOS—total oxidant status; FRAP—ferric reducing ability of plasma; GPx—glutathione peroxidase; SOD—superoxide dismutase; MDA—malondialdehyde.

**Table 4 antioxidants-11-01796-t004:** Selected biochemical components of bovine seminal plasma.

	Mean	SD	Min	Max
TP (g/L)	49.03	10.07	26.15	63.59
Alb (g/L)	16.69	4.16	10.84	26.43
Chol (mmol/L)	1.81	0.80	0.84	3.96
TAG (mmol/L)	1.10	1.08	0.19	4.50

TP—total proteins; Alb—albumins; Chol—cholesterol; TAG—triglycerides.

**Table 5 antioxidants-11-01796-t005:** Chemical elements detected in bovine seminal plasma.

	Mean	SD	Min	Max
Al (mg/kg)	0.80	0.01	0.79	0.81
Ba (mg/kg)	0.35	0.00	0.34	0.35
Ca (mg/kg)	330.32	72.85	159.43	470.98
Cd (mg/kg)	1.08	0.01	1.06	1.09
Co (µg/kg)	190.50	1.58	188.00	192.00
Cr (mg/kg)	1.25	0.01	1.24	1.27
Cu (mg/kg)	0.43	0.12	0.35	0.81
Fe (mg/kg)	2.46	0.17	2.11	2.82
Hg (µg/kg)	0.46	0.06	0.33	0.55
K (g/kg)	3.27	0.65	2.10	4.35
Mg (mg/kg)	80.43	35.96	40.39	175.49
Mn (µg/kg)	35.44	0.51	35.00	36.00
Mo (mg/kg)	1.07	0.03	1.02	1.13
Na (g/kg)	1.31	0.40	0.75	2.17
Ni (mg/kg)	2.50	0.58	1.49	3.87
Pb (mg/kg)	1.05	0.01	1.03	1.06
Sr (µg/kg)	188.28	1.64	185.00	190.00
Zn (mg/kg)	3.60	0.74	2.13	5.47

## Data Availability

All of the data is contained within the article and the Appendix A.

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
