# Peer review of "Sperm Quality Affected by Naturally Occurring Chemical Elements in Bull Seminal Plasma"

_antioxidants, 2022, doi:10.3390/antiox11091796_

Round 1

Reviewer 1 Report

My comments are included in the attached file

Author Response

Dear Reviewer.

Thank you for your effort and thorough revision of the current manuscript. Please see the attachment.

Reviewer 2 Report

The aim of this study is to evaluate concentrations of selected chemical elements, lipids, proteins in seminal plasma as well as its redox status.

Authors concluded that the obtained results suggest that seminal plasma composition could be used as a potential biomarker of bull reproductive health.

The topic of the study is interesting. However, the manuscript needs major revisions as outlined below:

1. Please modify the following sentence:

“Another important type of lipids, triglycerides, contribute to sperm energy production”.

The idea that, in the absence of glycolysable substrates, sperm might be able to oxidize fatty acids from endogenous phospholipids is well known in literature. Therefore, I suggest referring to “fatty acids” and not to “triglycerides”.

2. Results are mainly descriptive and possible molecular mechanisms for the observed effects are not investigatedAuthors should better clarify in the discussion section the link between sperm quality (motility), mitochondrial activity, redox homeostasis, and specific chemical concentrations in seminal plasma. 

Do chemical elements negatively affect mitochondrial function and, therefore, sperm quality? Are chemical elements cofactors of specific enzymes involved in sperm redox homeostasis? 

Authors should discuss this aspect.

3. Other suggestions:

- Has the relationship between seminal plasma cholesterol, triacylglycerols and proteins been investigated in other species?

- What is the role of seminal plasma cholesterol in sperm quality? In human seminal plasma it has been suggested that cholesterol may be important for spermatogenesis.

4. A paragraph including the limitations of the study should be added at the end of the manuscript.

Author Response

(The authors gave the same response as above.)

Round 2

Reviewer 2 Report

Authors modified the manuscript according to my comments.

Author Response

The collective of authors thanks the Reviewer for all the effort in revising the manuscript.